# Gelatin-Based Film as a Color Indicator in Food-Spoilage Observation: A Review

**DOI:** 10.3390/foods11233797

**Published:** 2022-11-24

**Authors:** Mannur Ismail Shaik, Muhammad Farid Azhari, Norizah Mhd Sarbon

**Affiliations:** Faculty of Fisheries and Food Science, Universiti Malaysia Terengganu, Kuala Nerus 21030, Terengganu, Malaysia

**Keywords:** food spoilage, color indicator, pH-sensitive film, food packaging, gelatin-based film

## Abstract

The color indicator can monitor the quality and safety of food products due to its sensitive nature toward various pH levels. A color indicator helps consumers monitor the freshness of food products since it is difficult for them to depend solely on their appearance. Thus, this review could provide alternative suggestions to solve the food-spoilage determination, especially for perishable food. Usually, food spoilage happens due to protein and lipid oxidation, enzymatic reaction, and microbial activity that will cause an alteration of the pH level. Due to their broad-spectrum properties, natural sources such as anthocyanin, curcumin, and betacyanin are commonly used in developing color indicators. They can also improve the gelatin-based film’s morphology and significant drawbacks. Incorporating natural colorants into the gelatin-based film can improve the film’s strength, gas-barrier properties, and water-vapor permeability and provide antioxidant and antimicrobial properties. Hence, the color indicator can be utilized as an effective tool to monitor and control the shelf life of packaged foods. Nevertheless, future studies should consider the determination of food-spoilage observation using natural colorants from betacyanin, chlorophyll, and carotenoids, as well as the determination of gas levels in food spoilage, especially carbon dioxide gas.

## 1. Introduction

Recently, there has been a great concern among consumers to detect the quality and safety of food products in the market due to the increasing awareness of high-quality and safe food, especially perishable food, such as seafood, poultry, and meat. Perishable food contains essential nutrients (water, glucose, and amino acids) required for bacteria such as *Pseudomonas* spp., *Enterobacteriaceae*, *Enterococcus* spp., lactic acid bacteria, and mesophilic total aerobic bacteria to grow [1]. The degradation of glucose by the bacteria is the main factor that causes off-flavor and off-odor in food products during storage. The spoiled food caused by the metabolic process can appear undesirable and unacceptable for consumers due to the alteration in sensory characteristics, i.e., texture, smell, taste, and appearance [2]. Conventionally, food-spoilage observation is undergone through sensory evaluation and chemical experiments [3]. In the last decade, scientists have been working on an alternative way to detect food spoilage by using a color indicator to monitor the microbial metabolites in food. It is helpful for food-spoilage observation to replace the conventional method, which can lead to human errors due to its inefficiency and low precision [4].

The use of color indicators plays a vital role in evaluating the freshness of the food product due to its sensitive nature toward various pH levels. It can offer significant advantages as it is low-cost, rapid, reliable, noninvasive, and non-destructive [5,6]. Usually, two sources of color indicators are present: synthetic sources (bromophenol blue, chlorophenol red, and alizarin) and natural sources (anthocyanin, curcumin, and chlorophyll). The criteria for the color indicator to be used in a food-spoilage observation is that the indicator must be easily distinguished by the naked eye, sensitive to various pH levels, and have a low response time between the appearance of distinctive color and change in pH [7]. During the decomposition of food by the microorganisms, the pH of the food can be altered due to the presence of total volatile basic nitrogen (TVB-N), fatty acids, ketones, etc. The mechanism of the color indicator has been proven to change its color due to the changes in pH when tested on the food product, as conducted in prior studies on beef [3,4], chicken and shrimp [8], and fish [9]. This color indicator tremendously impacts the packaging industry to improve food packaging and fulfill consumer needs. Recently, researchers have focused on developing color indicators from natural sources to detect food spoilage that may not adversely affect food. Food packaging is vital because it functions as a physical barrier between food and the environment to maintain the quality and prolong the shelf life of food products [10]. Biodegradable packaging is commonly used as food packaging to replace plastics from nonrenewable resources such as petrochemicals, which have some significant drawbacks due to being nonbiodegradable and raising critical environmental concerns among consumers [11]. Active packaging and smart packaging have significant contributions to the packaging industry due to the advantages of extending the shelf life of food products, food-spoilage observation, portable size, and low cost [8]. Active packaging is usually used to extend the shelf life of food products and improve the safety and quality of the food product. In contrast, smart packaging can monitor packaged food and indicate the product’s quality and freshness information for manufacturers, retailers, and consumers [12]. Commonly, the biopolymers used to immobilize the color pigment in functioning as smart packagings are protein-based film (chitosan, gelatin, and agar) and polysaccharide-based film (starch, carboxymethyl cellulose, and konjac glucomannan) [11,13].

Gelatin is a good biopolymer for film formation due to its film-forming ability, biocompatibility, and biodegradability, making it suitable for food packaging [14]. The gelatin-based film has been proven to protect against aroma, oxygen, and light. However, it has some significant drawbacks, such as poor barrier properties against water vapor and high moisture sensitivity [15]. Thus, it gives some negative feedback when the film is applied to food because it may dissolve, swell, or be integrated with water molecules [16]. The negative feedback of the gelatin-based film can be improved by incorporating natural extract and improving the functionalities, such as antioxidant and antimicrobial properties and color indicators [17]. The essential characteristic of a color indicator is that the film needs to be good in gas-barrier properties against carbon dioxide, oxygen gas, ammonia, and others for food-spoilage observation [18]. For instance, microorganism degradation from meat spoilage will release carbon dioxide, organic acids, and total volatile basic nitrogen (TVB-N), which will be detected by the color indicator and will change its color according to the pH inside the packaging [8]. Nowadays, consumers demand nontoxic, affordable, and environmentally possible products. The need for a pollution-free environment, the rise in crude oil prices, the demand of consumers for longer food shelf lives, and the proliferation of convenience packaging are contributing to the development of biodegradable packaging material for the market [19]. Consumers’ awareness of biodegradable packaging material is increasing as people are more educated. This sparks the interest of many researchers to produce safe, environmentally friendly, and affordable packaging material. Hence, research has increased on gelatin-based film as a biodegradable packaging material. Gelatin is the most widely used biodegradable packaging material because of the development of high-performance films [20]. Gelatin-based films can preserve, protect, or lengthen the shelf life of food products [21]. The gelatin-based film market in the U.S. is anticipated to rise rapidly during the anticipated period. In addition, the U.K., France, and Germany are expected to see high growth due to the wide range of uses for gelatin films in food. In contrast, China and India are anticipated to see high growth during the projected period due to the rapid adoption of new technologies and products in these countries [22]. Gelatin-based films are already produced widely in many developed countries. However, there is still no commercialized gelatin-based film in Malaysia, and the application of biodegradable films in packaging material is not yet widely applied.

This review would be able to provide alternative suggestions to solve the food-spoilage determination, especially for perishable food, by giving adequate exposure to the usage of the color indicators of the gelatin-based film. However, to date, no published review has been carried out on gelatin-based film as a color indicator in food-spoilage observation. Therefore, this review aims to critically investigate, discuss, and provide clear information about color indicators from gelatin-based films and natural colorants to observe food products’ quality and safety.

## 2. Food Spoilage

Food spoilage is leading, due to various mechanisms, which include microbial, chemical, and physical reactions, to rendering the food product undesirable or unacceptable for the consumer. Food spoilage mainly results from microbial growth and/or metabolism of spoilage bacteria, molds, and yeasts during storage [23]. Therefore, food spoiling from microbial contamination is advised to be thrown away.

### 2.1. Type of Food Spoilage

Food spoilages occur due to the transmission of infectious agents or pathogens, including bacteria, viruses, fungi, and protozoa, which can contaminate food products and cause foodborne illness in humans. The pathogens usually transmit to meat products due to preslaughter husbandry practices, animal age at the time of slaughtering, handling during slaughtering, evisceration and processing, temperature controls during slaughtering, processing and distribution, preservation method, type of packaging, and storage by the consumer [24]. Thus, the pathogens will cause damage to the food products, which is known as microbial spoilage. There is previous research conducted by various researchers who found that the specific microorganisms that induce meat spoilage in poultry meat are *Pseudomonas, Enterobacteriaceae*, lactic acid bacteria, and *Brochothrix thermosphacta* [25,26]; in lamb meat are *Pseudomonas*, *Shewanella putrefaciens, Brochothrix thermosphacta, Enterobacteriaceae*, lactic acid bacteria, and *Clostridium* spp. [27]; and in beef meat is *Carnobacterium, Brochothrix*, and *Leuconostoc* spp. [28]. Therefore, microbial spoilage is one type of food spoilage involving lactic acid bacteria (LAB), spore-forming bacteria, and Gram-negative bacteria.

Common genera of LAB include *Lactobacillus, Weisella, Leuconostoc, Lactococcus, Pediococcus, Streptococcus*, and *Enterococcus*. LAB, also known as facultative anaerobic bacteria (tolerating the presence or absence of oxygen), are usually involved in the metabolic activity of food and can cause undesirable spoilage to foods, such as greening and slime production to meat, off-flavor, bloating, or exploding of vacuum-sealed pouches, and gas formation [29]. For example, it is reported that the presence of *Lactobacillus curvatus, Lactobacillus sakei*, and *Leunostoc mesenteroides* can lead to carbon dioxide production in the sliced vacuum-packed cooked ham, causing the packaging to lose the vacuum condition [30]. In addition, LAB can also cause off-odors (pungent, fishy, rotten egg, and ammonia-like) and slime production due to the secretion of exopolysaccharides by the bacteria on refrigerated chicken meat, as reported by Katiyo et al. [31].

Spore-forming bacteria such as *Alicyclobacillus* spp., *Bacillus* spp., and *Clostridium* spp. are categorized as heat-resistant bacteria because they survive at high temperatures or pasteurization processes. *Clostridium tagluense, Clostridium putrefaciens, Clostridium algidicarnis, Clostridium frigoris*, and *Clostridium gasigenes* are detected on spoiled meat, while *Clostridium estertheticum* is detected on vacuum-packed chilled stored meat [32]. A study has reported that Dezhou-braised chicken treated at 121 °C for 30 min shows positive effects of maintaining fresh odor and texture and improving the shelf-life compared to Dezhou-braised chicken treated at 84 °C for 35 min due to the presence of *Clostridium* spp. and *Bacillus* spp. [33]. Thus, it is proved that the spore-forming bacteria cannot withstand sterilization temperatures. In addition, Gram-negative bacteria, including *Acinetobacter, Aeromonas, Pseudomonas, Salmonella, Enterobacteria, Alcaligenes, Alteromonas, Flavobacterium, Moraxella*, and *Archromobacter*, can contribute to the spoilage of poultry, pork, lamb, mutton, and beef [34]. Gram-negative bacteria play an important role in producing biogenic amines such as putrescine. Putrescine is considered carcinogenic because it can contribute to undesirable toxicological effects when consuming food in an excessive amount, such as headaches, hypertension, low blood pressure, edema, etc. Putrescine production happens in three metabolic pathways: direct decarboxylation of ornithine, biosynthetic, and biodegradation of arginine. The last two pathways usually occur in the presence of *Pseudomonas* and *Enterobacteria* with the catalyzation of indigenous enzymes such as arginine decarboxylase and agmatinase [35].

Subsequently, enzymatic spoilage also contributes to the type of food spoilage that involves hydrolases (involving water) and oxidoreductases (involving oxygen) in food, resulting in hydrolytic reactions and oxidation-reduction reactions by breaking the chemical bond in the food composition [29]. Hydrolase enzymes, such as α-amylases, β-amylases, and glucoamylases, can hydrolyze the α-1,4-glycosidic bond, β-1,4-glycosidic bond, and both (α-1,4- and β-1,4-glycosidic bond) of meat glycogen, respectively, to produce simpler molecules such as glucose, maltose, dextrin, and maltotriose. The presence of glucose contributes to food spoilage because it is an essential energy source for the growth of microorganisms [36]. In addition, oxidoreductase enzymes such as lipoxygenases can catalyze lipid oxidation, leading to oxidative rancidity in lipid-containing food [37]. The activity of enzymes is a natural process that leads to meat deterioration (known as enzymatic spoilage) because the enzymes remain active and participate in biochemical changes in the muscles of meat products during and after slaughter [1]. In addition, the enzymes present naturally in food or produced by microorganisms can act as a catalyst for the breakdown of complex food composition (carbohydrates, fats, and proteins) into simpler forms, resulting in softening and greenish discoloration in meat [24]. It is also reported that the protease enzyme is involved in the proteolytic activity to produce TVB-N, which results in off-odors of meat products [1].

### 2.2. Factors Affecting Food Spoilage

#### 2.2.1. Extrinsic Factor

The extrinsic factor depends on the external environment of the food products that affect the growth of the microorganisms. The parameters involved are storage temperature, moisture, and atmosphere conditions (aerobic, anaerobic, and modified atmosphere packaging) [29]. Understanding time and temperature management to control food spoilage are essential to improve the shelf-life of food products because they are the main factors contributing to the growth of microorganisms. The growth of microorganisms depends on the survivability of the microbes (optimum temperature) that are classified into psychrotroph (2–7 °C), mesophile (10–40 °C), and thermophile (43–66 °C) [29,38]. At below or above optimum temperature, it will retard the microorganisms from multiplying. A study on fish meat (sturgeon fillet) by Li et al. [39] found that only *Acinetobacter johnsonii* can multiply at chilled temperatures (4 °C) compared to *Shewanella baltica, Pseudomonas fragile, Pseudomonas koreensis*, and *Pseudomonas Antarctica*. Thus, it is proven that bacteria can only survive by depending on the survivability of the microbes. The gaseous atmosphere within the packaging is also crucial in maintaining the quality of the food and retard the growth of microbial spoilage. Aerobic bacteria such as *Pseudomonas* spp. (*P. fragi, P. ludensis*, and *P. putida*), *Acinetobacter* spp., and *Moraxella* spp., are the common bacterial spoilage in aerobically packaged meat within the temperature range from −1 °C to 25 °C, while anaerobic microbes such as *Shewanella* spp., *Brochothrix* spp. (*B. thermosphacta* and *B. Compestris*), *Serratia* spp. and lactic acid bacteria species will become dominant and retard the aerobic bacteria from overgrowth in vacuum-packed meat [29,40].

#### 2.2.2. Intrinsic Factor

The intrinsic factor depends on the internal environment of the food products that affect the growth of microorganisms. The parameters involved are the physical and chemical composition of the substrate, water activity, pH, nutrient availability, initial microflora, and the presence of natural antimicrobial substances [29]. Meat muscles are rich in proteins, lipids, minerals, and vitamins, which are essential energy sources for the microorganisms present in meat [17]. Usually, the pH of the meat after slaughter is slightly lower (pH 5.4–5.8) due to the presence of lactic acid from the hydrolysis of glycogen during anaerobic glycolysis, which inhibits the growth of microbes. During storage, the pH will increase to a pH of 6.0, which leads the microbes to multiply and results in spoilage [29]. Water activity is vital in providing a favorable environment for microbial growth in meat. Bacteria can grow at a higher water activity (from 0.75 to 1.0) than mold and yeast (0.62). It is also reported that reducing water activity from 0.99 to 0.97 can inhibit the growth of *Pseudomonas* and *Enterobacteriaceae* in meat [41].

### 2.3. Characteristics of Spoiled Food

The characteristics of spoiled food are discoloration, slime production, off-odor, and off-flavor. Discoloration occurs due to the growth of lipolytic yeast, such as *Saccharomycopsis lipolytica*, which can cause brown-black spots on meat fat. The bacterial production of hydrogen sulfide, which derives from cysteine, happens due to the limitation of glucose and oxygen, which can cause the conversion of muscle pigment to sulfmyoglobin (green pigment). Hydrogen peroxide oxidation of nitroso hemochrome to Chloe myoglobin also forms green spots on meat [29]. Lactic acid bacteria, *Leuconostoc* spp., *Enterococcus* spp., *Pediococcus* spp., and *Carnobacterium* spp., synthesize the production of hydrogen peroxide. (*C. divergens*), and *Weisella* spp. (*W. viridescens*). Next, slime production on the meat surface is due to *Leuconostoc*, lactic acid bacteria, *Carno-bacterium*, and *Weisella* during cold storage [40]. Moreover, the presence of esters, ketones, aldehydes, hydrocarbons, alcohols, benzenoids, terpenoids, nitrogen–sulfur compounds, amines, and volatile fatty acids are the main factor that causes off-odor and off-flavor in the meat. Lactic acid bacteria and *B. thermosphacta* can metabolize glucose in aerobic conditions and ribose, glycerol, and amino acids in anaerobic conditions, leading to the volatile compound’s production. The presence of volatile compounds will result in sour smells and sour flavors. In contrast, a rotten-egg smell is produced due to the presence of *S. putrefaciens, S. liguefaciens, E. agglomerans*, and *H. alvei* due to hydrogen sulfide production [42].

## 3. Color Indicator for Food Spoilage

### 3.1. Introduction to Color Indicators in Food

A color indicator is a new technology for the real-time monitoring of freshness to notify the level of food spoilage that helps consumers to choose the best quality food products [4]. Conventionally, the identification of food spoilage detected by a chemical experiment, which involves the determination of total volatile basic nitrogen (TVB-N), triphenyl tetrazolium chloride (TTC), and biological methods (total viable counts, TVC), as well as lactic acid bacteria, *Brochothrix thermosphacta*, *Pseudomonas* spp., and *Enterobacteriaceae* determination [4]. In addition, food spoilage is also determined by changes in sensory characteristics such as color, texture, smell, and appearance of the food products [3]. Sometimes, the uses of the conventional methods to identify food spoilage are rejected due to the inefficiency and low precision, because they lead to human errors, and because consumers have difficulty differentiating the product quality by simply relying on sensory characteristics such as color [4]. For example, fresh and spoiled meat colors have an insignificant difference; thus, it is difficult for consumers to choose the desired quality to fulfill their needs [1]. Furthermore, it is also reported that the biological method using Fourier transform infrared spectroscopy to monitor spoilage on minced beef is high cost, only relies on trained panelists, is lengthy, and is not able to give the immediate answer required compared to the color indicator [43]. Furthermore, the color indicator needs to meet several criteria for freshness monitoring, e.g., easy to distinguish the color changes with the naked eye, sensitivity towards various pH levels, and low response time between the appearance of distinctive color and changes in pH [44]. Different types of color indicators in food packaging that are available in the market are shown in Figure 1. Commonly, the color indicator has two essential parts, i.e., solid support and colorants [7].

The function of the solid support is to immobilize the colorants, which usually use natural biopolymers (biopolymer-based film), such as polysaccharides and proteins [46]. However, the biopolymer-based film can be easily degraded by microorganisms and other volatile compounds in the packaging environment. Therefore, combinations with other polymers have been introduced, i.e., chitosan–starch [9], starch–gelatin [8,47], starch–polyvinyl alcohol [48], starch–nanomaterial [49,50], and carboxymethyl cellulose–polyvinyl alcohol [51], and combinations with colorants [3,6,52] to overcome the negative feedbacks of the packaging film. Moreover, the colorants are the main character as a color indicator that undergoes color changes to evaluate the quality of food products by detecting the changes in pH, gas level, and microbial activity in the packaging [53]. Synthetic and natural sources are two types of colorants commonly used in developing color indicators.

### 3.2. Type of Color Indicator

#### 3.2.1. Synthetic Sources

Synthetic colorants, also known as acid-base indicators, are affected by a wide range of intermolecular interactions (Bronsted and Lewis acid-base, hydrogen bonding, di-polar, and π-π interactions) between the analyte and colorants [54]. The critical element for synthetic dyes to be used as a color indicator is the ability to change their color in an acidic or basic environment. Various types of artificial colorants are commonly used as food spoilage indicators, such as tetraphenylethylene (TPE), bromocresol green, methyl red, xylenol blue, crystal violet lactone, bromophenol blue, cresol red, Bromo-cresol purple, bromothymol blue sodium salt, alizarin, phenol red, chlorophenol red, and rosolic acid, that can indicate meat spoilage during bacterial decomposition due to the ability to change color at a different level of pH [55,56]. A study by Liu et al. [57] found that TPE dye that is immobilized to polyaniline polymer (PANI) will change its color from emerald-green to peacock-blue when exposed to the spoiled fish due to the releasing of volatile amines (triethylamine, trimethylamine, and ammonia). In addition, bromocresol green dye is also used as a colorimetric sensor to indicate fish spoilage. It is reported that the bromocresol green turns yellow-green from yellow in four hours, then turns green in ten hours, and finally turns dark green in 22 h due to the increase of TVB-N in fish from 10 mg/100 g to 23.11 ± 2.22 mg/100 g. The color will turn dark blue when TVB-N exceeds 25 mg/100 g [58]. However, studies using other sources of synthetic colorants are still lacking.

#### 3.2.2. Natural Sources

Colorants from natural sources have been suggested as a suitable alternative to be used as a color indicator over synthetic colorants in food packaging due to their obvious side effects, toxicity in the short and long term, possible carcinogenic effects, and allergic reactions, especially in children (ADHD-like hyperactivity), and their ability to disrupt biological systems and environment [59]. Natural colorants can be used as an indicator to detect food spoilage due to their ability to see changes in pH, gas level, and microbial activity in the packaging [53]. So far, a color indicator that detects changes in gas levels, especially carbon dioxide, is still lacking. Natural colorants are usually sensitive to various pH levels, which can change their color according to the pH. So far, a color indicator that detects changes in gas levels, especially carbon dioxide, is still lacking. Natural colorants are usually sensitive to various pH levels and can change their color according to the pH, as shown in Figure 2. In addition, the colorants immobilized in the polymer matrix will change color due to the rise of volatile amines by pathogenic microbes in spoiled food [60]. This positive impact of natural colorants has prompted various researchers to develop color indicators due to the excellent potential for the development of real-time monitoring of freshness from natural sources such as anthocyanin, curcumin, chlorophyll, carotenoid, and betacyanin [52,61].

Anthocyanin is a natural phenolic indicator dye that can be extracted from several sources, such as red cabbage, blueberry, black soybean, raspberry, and purple cauliflower. It is soluble in water and appears red, pink, purple, green-yellow, and colorless depending on the pH [62]. There are various anthocyanin types, such as cyanidin, pelargonidin, peonidin, delphinidin, malvidin, and petunidin, consisting of 3-glucoside of the aglycone anthocyanidin, which is the sugar-free analog of anthocyanins [59]. The changes in the color of anthocyanin at different pH levels are due to the mechanism reaction of cations, such as flavylium cation (red pigment) that appears at strong acid conditions at pH < 4, carbinol and pseudo-base chalcone (colorless dye) appear at pH 4–6, quinoidal anhydrous-base (blue/purple pigment) appears at pH 6–8. Colorless chalcone (light-yellow pigment) appears at pH > 8 [63]. In addition, the substitution configuration of anthocyanin also leads to the color changes of anthocyanin. For example, increasing methoxy groups will generate a reddish color, while hydroxy groups will generate bluish color [59]. Incorporating anthocyanin as a natural additive to the pH-sensitive film becomes an innovative tool for the indication of real-time monitoring of freshness [61]. Curcumin can be found in Curcuma longa, which is known as a bright yellow pigment due to the presence of curcuminoid pigment. The curcuminoid pigment consist of three types of curcumin, which are curcumin I (curcumin), curcumin II (desmeth-oxy-curcumin), and curcumin III (bisdemethoxycurcumin) [64]. Curcumin has poor solubility in water and hydrocarbon solvents but is very soluble in polar solvents such as ethanol. Curcumin can undergo hydrolytic degradation due to its instability at high temperatures (>190 °C). The color stability of curcumin in an aqueous solution depends on the pH of the solution, where the optimum pH is 1–6. The color change from yellow to red occurs at a pH lower than pH 3 and is more significant than pH 7 [65]. The color changes of curcumin at different pHs give the ability to indicate the quality of food since pH changes occur resulting from food spoilage [18]. Nonetheless, there is also a lack of studies using natural colorants from chlorophyll, carotenoid, and betacyanin.

## 4. Food Packaging

Food packaging is critical in containment, protection, communication, and convenience to prevent physical, chemical, and biological damage and provide product information [66]. There are various types of food packaging present in the market that is innovated to offer more convenience to consumers and also to meet consumers’ lifestyles. Usually, food packaging acts as passive protection for food products, providing a protective barrier between food and environmental influences (oxygen, moisture, light, dust, pest, and volatiles) [66]. Food packaging technology significantly impacts food products to minimize the possible deterioration or degradation of quality and improve the shelf life of packaged food [67]. Scientists have worked hard to enhance packaging by developing various technologies for food packaging, such as vacuum packaging, modified atmosphere packaging (MAP), active packaging, and smart packaging.

### 4.1. Active Packaging

Active packaging is commonly used to extend the shelf life of food products, improve safety, and maintain the quality of the product. The active packaging usually performs monolayered by incorporating an active compound with the polymer and a multilayered system by entrapping the active compound between the polymer layer [68,69]. A monolayered film can be used if the film has good antioxidant packaging requirements, such as transparency and good sealing condition. Still, this film has disadvantages in controlling the active compound from loss to the environment. Thus, the multilayered film is used to minimize the migration of active compounds to the environment because the active layer is protected with a protective layer [70]. The development of active packaging film can be classified into three methods, which are the casting method (by mixing the active compound and polymer matrix into an appropriate solvent), the extrusion method (by incorporating the active compound into the melted polymer), and the coating method (by treating the active compound with a physical or chemical process to favor the adhesion of the active compound) [71].

Biopolymer or nonsynthetic polymers, such as proteins, polysaccharides, and lipids, are widely used in film packaging to replace or minimize the use of synthetic polymers, i.e., polycarbonate (PC), low-density polyethylene (LDPE), high-density polyethylene (HDPE), polystyrene (PS), polyethylene terephthalate (PET), and polypropylene (PP) [72]. This is because synthetic polymer has disadvantages, such as being nonbiodegradable, challenging to recycle, and can cause environmental damage (air and water pollution) [73,74]. Various sources of proteins, polysaccharides, and lipids have been used for the development of active packaging film to improve the quality and shelf life of meat products such as gelatin [75], starch [76,77], chitosan [78], carboxymethyl cellulose [79,80], and propolis extract [81]. The use of biopolymer in developing active packaging film should be good in mechanical and physical properties and have good properties of sealing, hermeticity, barrier, etc. [70]. Plasticizers (sorbitol, glycerol, polyethylene glycol, etc.) are incorporated into the polymer due to the drawbacks of the polymers, such as being too brittle and rigid, which leads to limitations for application in the food industry. The plasticizers will react with the polymer’s chain, increasing flexibility and improving mechanical and physical properties [82]. Moreover, active packaging can absorb/scavenge properties and release/emit properties [67]. The scavenging system of active packaging is responsible for absorbing volatile compounds that can cause off-flavor, oxidation, and discoloration to the food product by incorporating absorbing materials. For example, a study by Byun et al. [83] found that the incorporation of oxygen scavenger (Iron (II) chloride) into gelatin film can reduce the oxygen level of active packaging from 20.90% to 4.56% after 50 days of storage. In addition, the releasing system of active packaging can provide various advantages. For example, a smaller number of antioxidants is needed than direct addition to the food product and can indirectly produce a no-preservative food product. Moreover, the antioxidant activity can also be centered on the more sensitive product surface through the migration from the packaging to the food matrix [84]. It has been proved that the incorporation of antioxidants can give benefits to the food product, especially meat products (beef, foal, pork, and turkey) such as polyvinylpolypyrrolidone washing solution (PVPP-WS) [85], citric acid [86], rosemary extract [85], green tea extract [87], green, black and oolong tea extract [73], oregano essential oil [87], encapsulated green tea extract [88], citrus extract [89], and cinnamon + rosemary [90] to reduce lipid and protein oxidation, prevent discoloration, and improve sensory characteristics of the meat product.

### 4.2. Smart Packaging

Smart packaging is a type of innovative packaging that is extended from traditional packaging. Traditional packaging aims to prevent the products from leaking, breaking, or being contaminated by pathogenic microbes; to communicate to the consumers the information about the food products (nutritional content or cooking instruction); to provide convenience; and to provide containment during transportation [91]. The presence of smart packaging may accommodate and fulfill consumer needs because traditional packaging is no longer sufficient (increasing awareness of biodegradable packaging, increasing product complexity, and minimizing the carbon footprints of manufactured products) by enhancing the functionality of traditional packaging [92]. In addition, smart packaging can provide product quality information due to microbial growth or chemical changes in the food product inside the packaging. The smart packaging system can also be classified as direct and passive indicators attached to the packaging labels, incorporated into a polymer matrix, or printed onto food packaging [12,93].

Moreover, smart packaging can monitor the quality of food products due to the presence of sensors inside the packaging to sense the freshness of food products, which, in turn, also shows whether the product is safe to consume. Usually, pH-sensitive dyes or colorants from natural or synthetic sources are used as a sensor that is entrapped within the polymer matrix (e.g., gelatin, starch, agar, tara gum, cellulose, and k-carrageenan) that will change its color when alteration of the pH occurs due to the presence of volatile compounds from food spoilage [94]. The reaction of the pH sensor can be classified into three types: firstly, the physical adsorption of immobilized colorants on a solid support, such as an ion exchanger. Secondly, the colorants, such as cellulose polymer, are covalently attached to a hydrophilic support on the packaging. Thirdly, the colorants are physically entrapped in a polymer matrix [95]. Natural pigments replace synthetic colors (e.g., bromocresol, methyl red, and chlorophenol) because they are safer and eco-friendlier. Various types of bioactive compounds from natural sources have been incorporated into the smart packaging by the researchers, such as potatoes/sweet potatoes [96], purple potatoes [97], red cabbage [98], roselle calyx [48], turmeric [99], and cactus pears [100].

## 5. Gelatin-Based Film

### 5.1. Sources of Gelatin

Gelatin is widely used in the food industry due to its ability to serve as a thickener, emulsifier, plasticizer, binding agent, etc., that is extracted from animals such as pigskin (46%), bovine hides (29.4%), bovine bones (23.1%), and other sources (1.5%), which consist of type A and type B gelatin [101]. Type A and type B gelatin are two types of gelatins that are produced based on the pretreatment, which is acid-treatment gelatin (the isoelectric point from pH 6 to 9), found in pigskin, and alkaline-treatment gelatin (the isoelectric point at pH 5), found in bovine hides, respectively [102]. A study by Aramwit et al. [103] reported that type B gelatin contains a higher cross-linking degree and physicochemical properties than type A gelatin, leading to a slower degradation rate. Thus, it shows that the gelatin’s quality depends on its source. In addition, the source of the gelatin is also essential for religious concerns because Muslims and Jews are prohibited from consuming pork-derived products. At the same time, Hindus are forbidden from consuming bovine products [104]. Furthermore, the source of gelatin also contributes to the quality of the gelatin film. For example, a comparative study by Nur Hanani et al. [105] showed that gelatin-based film from fish had the lowest water-vapor permeability compared to the gelatin-based film from beef and pork because fish-derived gelatin consists of low proline and hydroxyproline content.

### 5.2. Properties of Gelatin-Based Film

Gelatin can be used as an edible biopolymer for food packaging due to several advantages, such as high biocompatibility, good film-forming ability, and biodegradability. In addition, the presence of α and β chains in gelatin gives different gel strengths. It is reported that more α chain in the gelatin leads to higher gel strength [106]. A mixture of the gelatin-based film with other polymers or active biochemical reagents can strengthen the film to improve the extensibility, dispensability, flexibility, elasticity, and rigidity due to some significant drawbacks of the film, such as ease of dissolving in water, coupled with inferior thermal stability, poor tensile strength, poor water-vapor permeability, and poor mechanical properties [15,107]. It is reported that incorporating other materials into the gelatin-based film can increase the film polymer’s cross-link by hydrogen bond formation between the polymer and the added material [94]. In a study by Wang et al. [107], adding 10% of aldehyde starch crystal can improve the thermal stability of gelatin-based film and maximize the tensile strength, elongation at break, contact angle, and barrier efficiency.

## 6. Characterization of Gelatin-Based Film towards Color Indicator in Food Packaging

### 6.1. Color Stability

Color stability indicates the ability of the film’s color to retain and prevent discoloration or loss of color. It is because the film’s color is the main characteristic that acts as a color indicator for food-spoilage observation [53]. The color loss of the film occurs due to the degradation of natural extracts such as anthocyanin and curcumin. Various factors can contribute to the degradation of the natural extract, including light quality, oxygen availability, heat, pH changes, structure and concentration of pigments, presence of co-pigments, sugars, metal ions, enzymes, and their degradation products such as sulfur dioxide [8,66]. Anthocyanin from red cabbage, black carrots, red radishes, red potatoes, red corn, beans, roselle, black/purple rice, and curcumin from turmeric are often used as a color source for the development of film packaging as a color indicator [108]. The effect of color stability using various sources of natural extract for the application of film packaging as a color indicator is shown in Table 1.

A comparative study by Prietto et al. [6] reported that the color indicator incorporated with anthocyanin from black beans showed low instability compared to the red cabbage at room temperature, with the incidence of light for 28 days due to the acylation of anthocyanin. The hydroxyl group of anthocyanin is commonly acylated with organic and inorganic acids (sulfuric acid), affecting anthocyanins’ chemical stability. The presence of glucoside anthocyanin in red cabbage also leads to the stabilization of anthocyanin compared to the monoglucoside in black beans due to the possibility of forming unstable intermediates that can be degraded into phenolic acids and aldehydes. A study by Chayavanich et al. [8] showed that color indicators from the gelatin-based film incorporated with anthocyanin are more stable when stored at refrigeration temperature compared to room temperature. It is because the concentration of anthocyanin decreased gradually over time at all temperatures and more rapidly at higher temperatures. For instance, after 60 days, 11% of delphinidin was lost at 4 °C compared to 99% at 37 °C, while 17% of cyanidin was lost at 4 °C compared to 98% at 37 °C [113]. A comparative study between anthocyanin and curcumin by Chen et al. [109] showed that the color stability of color indicators containing anthocyanin is poorer than curcumin due to anthocyanin degradation during long-term storage. Another study by Peralta et al. [114] showed that the natural polymeric films developed from different combinations of aqueous hibiscus extract (HAE) with gelatin, starch, and chitosan. Anthocyanin, present in HAE, changes film color at various pH levels.

### 6.2. Color Response towards Various pH

The color response is an indicator to determine the color changes of the gelatin-based film incorporated with natural extracts such as red radish, purple sweet potato, roselle, blackberry, black/purple rice, and turmeric towards various pH levels [8]. Therefore, the gelatin-based film containing natural extract needs to be sensitive to multiple pH levels and have a low response time between the appearance of distinctive color and changes in pH to be used as a color indicator [44]. Usually, the color response of the gelatin-based film is determined by immersing the film in different pH levels (pH 2–12) and using a colorimeter to determine the total color difference [61]. Additionally, volatile ammonia is used to determine the film’s sensitivity towards volatile nitrogen compounds because the compounds are generally produced during meat spoilage [50,109]. A study by Chayavanich et al. [8] reported that incorporating the gelatin-based film with anthocyanin extract from red radishes shows a visible color ranging from orange to pink to blue and yellow at different pH levels. A comparative study between purple sweet potato extract (PSPE) and red cabbage extract (RCE) by Zhang et al. [115] showed that PSPE has bolder color than RCE due to abundant anthocyanin content; thus, it is easy to distinguish the color difference between the film from pink (pH 2–6) to light purple (pH 7–8), and to dark blue (pH 9–12). The comparison of sensitivity towards ammonia gas between anthocyanin and curcumin reported that anthocyanin shows high response sensitivity in terms of color difference compared to curcumin; thus, it indicates that anthocyanin is more sensitive toward ammonia [109]. Furthermore, it reported that a higher concentration of curcumin makes the color of the gelatin-based film brighter, which is caused by the yellow color of curcumin; thus, the color indicator can change in a way that is noticeable to the naked eye under acidic and alkaline solution conditions [116]. Table 2 shows the pH range of film packaging as a color indicator from various sources of natural extract and application on meat-spoilage observation.

### 6.3. Fourier Transform Infrared (FTIR) Spectrum

The gelatin-based film’s Fourier transform infrared (FTIR) spectrum was analyzed using an FTIR spectrometer to observe the morphology of the film’s matrix [50]. From the FTIR spectrum, the level of the functional group of the film’s matrixes can be detected, such as a decrease in the OH group due to the formation of a hydrogen bond between the OH group from the biopolymer and natural extract. As a result, the hydrogen bond in the film matrix will increase, leading to changes in the absorption peak of the FTIR spectrum [119]. The absorption peak of the film packaging as a color indicator at different concentrations of the extract is shown in Table 3. In addition, stretching vibration of the carbon-carbon double bond (C=C) of the aromatic ring from natural extract due to the interaction of anthocyanin and polymer matrix can also be detected, leading to the stronger chemical and physical structure of the film [115].

Furthermore, increasing N-H bonding can be detected due to the interaction between the amino group of biopolymers and the O-H group of natural extract, resulting in stronger intermolecular interaction of the film. A study by Liu et al. [123] reported that the incorporation of haskap berry extract (HBE) in the gelatin-based film could broaden and intensify the amide-A, N-H, and O-H stretching due to the formation of intermolecular hydrogen bonding between the hydroxyl group in HBE and gelatin, thus resulting in a dense network of the film matrix. It was also reported that incorporating red radish extract into the gelatin-based film can increase the C-H, C-O, and C-C bending vibration, C-O-C glycosidic bond, and C=O stretching of amide due to the chemical interaction between the polymers and the extract [8].

### 6.4. Water-Vapor Permeability

The water-vapor permeability of the film indicates the potential of water permeability or diffusivity and solubility of the water transfer through a film [72]. The determination of water-vapor permeability plays an essential role in the production of the gelatin-based film to function as a good film material and as a color indicator because water molecules are a transferable element that can pass through the film (via adsorption or desorption), which leads to degradation of the film properties. Good properties of the film generally have low water-vapor permeability due to its highly sealing effect that leads to the water molecules that have difficulties transferring through the film [125]. Several factors can affect water-vapor permeability, including temperature, relative humidity, food type, water activity, and film characteristics [126]. In addition, the concentration of natural extract incorporated into the gelatin-based film can affect the water-vapor permeability of the film by interfering with the morphology of the film’s matrix [61]. A study by Zhang et al. [115] reported that the incorporation of anthocyanin from purple sweet potato extract increased the film’s water-vapor permeability compared to red cabbage extract. This is because purple sweet potato contains abundant anthocyanin pigment compared to red cabbage with the same concentration. This leads to the disruption of the molecular structure, resulting in a porous surface of the film. It also reported that the incorporation of black soybean seed coat extract from 5% to 15% of *w*/*w* could gradually decrease the water-vapor permeability of the film packaging due to the formation of a dense network [127]. In addition, the uses of anthocyanin from mulberry pomace extracts can also enhance the water-vapor barrier properties of the film by reducing the water-vapor permeability due to the interaction of the phenolic group of the extract with the hydroxyl group of the film to form a complex structure [17]. The details of the findings on the relationship between the concentration of natural extract and water-vapor permeability of gelatin-based film are shown in Table 4.

### 6.5. Antioxidant and Antimicrobial Properties

The presence of polyphenols (flavonoids, phenolic acids, stilbenes, and lignans) in the natural extract contributes to the red, purple, and blue color of flowers, vegetables, and fruits has the potential as antioxidant and antimicrobial properties [88]. Recently, the utilization of natural extract for the development of color indicators from gelatin-based films, such as anthocyanin, has gained increasing attention due to the ability to act as an antioxidant and antimicrobial potential, which can indirectly maintain stability and extend the shelf life of the food product [122,131]. Therefore, apart from the fact that the color indicator can give quality information, it can also maintain the quality of the food product. Usually, the antioxidant of the film is determined by the 2,2-diphenyl-1-picrylhydrazyl (DPPH) assay or the ferric reducing antioxidant power (FRAP) assay [61]. For the determination of antioxidants, the DPPH assay (deep-violet color) will react with the antioxidant or any radical species (R^−^) by providing electrons and reducing to DPPH-H or DPPH-R (colorless or pale-yellow color) [132].

On the other hand, the FRAP assay is used to measure the reduction of Fe^3+^ (colorless) to Fe^2+^ (intense blue color) by the electron-donating antioxidant [133]. In addition, the determination of antimicrobial properties uses the agar diffusion method to detect the presence of foodborne pathogens (*E. coli, Salmonella, S. aureus*, and *L. monocytogenes*) [124]. In a study by Liu et al. [123], it was reported that the scavenging activity of the gelatin-based film increased with the increase in haskap berry extract concentration due to the releasing of polyphenols, resulting in the prevention of oxidative damage when applied to the food products. The presence of catechins from green tea extract with a 20% concentration shows the highest radical scavenging activities compared to the Pu-erh tea extract [134]. The addition of butterfly extract also indicates a significantly increased antioxidant activity of gelatin film due to the potent antioxidant activity of phenolic flavanols of anthocyanin [128]. Incorporating curcumin into the gelatin film can also improve the film’s antioxidant properties as a scavenging activity, which is significant in preserving food. Furthermore, it can also enhance the antibacterial properties against Gram-positive (*E. coli*) and Gram-negative bacteria (*S. aureus*) by reducing the growth rate from 701 to 11 CFU/mL and 342 to 79 CFU/mL, respectively [116]. The effect of the natural extract on the antimicrobial and antioxidant properties of film packaging as a color indicator is shown in Table 5.

## 7. Research Gap between Gelatin-Based Film and Paper as A Color Indicator

A solid matrix such as polymer and paper is a significant component for the color indicator to immobilize the colorants. Biopolymers, such as gelatin-based films, are preferred as a solid matrix due to their positive impact on the environment, including biodegradability, renewability, and eco-friendliness, as well as having good volatile gas binding ability and a faster release of colorants. Although the gelatin-based film has drawbacks (high water solubility and low water-vapor barrier properties), a combination of the gelatin-based film with other polymers, such as starch, agar, carrageenan, etc., can solve the problems [128,136]. Besides that, incorporating natural extracts, such as anthocyanin and curcumin, can improve the film’s drawbacks and act as a new indicator of freshness [119,120]. In addition, the paper contributes to the color indicator by immobilizing the colorants into the paper using a spin-coated method or by simply immersing it in the colorants. A study on color indicators by Husin et al. [3] reported that the incorporation of butterfly pea extract into the filter paper using a spin-coated method was able to monitor the freshness of beef. The use of synthetic colorants is also able to serve as a freshness indicator. A study by Hidayat et al. [137] reported that immersing the filter paper in 10 mL of phenol red and bromothymol blue for 12 h was able to indicate the quality of fresh meat. However, since the intermolecular forces between the paper polymer are strong, it is difficult for the colorants to bind to the paper. As a result, the dyes easily leach out, leading to the color indicator’s low effectiveness. Meanwhile, synthetic colorants, such as phenol red and bromothymol blue, are not suitable for food products due to their toxicity to human health [59].

## 8. Future Trends

Recently, there has been an increase in demand for the development of packaging that can indicate the quality and safety of food products, especially perishable food, including poultry, beef, and seafood. Scientists have been working on alternative ways to develop color indicators for detecting food spoilage. The color indicator can detect food spoilage due to its sensitive nature toward the various pH levels by incorporating colorants from synthetic or natural sources and can detect gas levels such as carbon dioxide, ammonia, and oxygen in food-spoilage observation. The spoiled food will release total volatile basic nitrogen (TVB-N) that will interfere with the pH in the packaging environment. As a result, the color indicator will change according to the pH inside the packaging. However, some limitations exist in developing color indicators, such as toxicity and carcinogenicity of synthetic colorants compared to natural colorants. In addition, there is also a lack of studies for the determination of meat spoilage observation using natural colorants from betacyanin, chlorophyll, and carotenoids, as well as the determination of gas levels in food spoilage, especially carbon dioxide gas. Therefore, future development needs to include the development of color indicators using betacyanin, chlorophyll, and carotenoids, as well as the determination of gas levels, especially carbon dioxide gas.

## 9. Conclusions

In conclusion, as a color indicator in food-spoilage observation, the gelatin-based film can be obtained by incorporating bioactive compounds from natural extract. This review found that adding the crude extract to the gelatin film improved the physical and mechanical properties of the film, such as morphology, color stability, and water-vapor permeability. At the same time, it provides antioxidant and antimicrobial properties to the film packaging, which indirectly can maintain and extend the shelf life of the food product. Furthermore, the quality and safety of food products can be monitored through the film’s sensitive nature towards various pH levels in detecting food spoilage by color indicator. The color indicator can detect the pH changes due to the presence of synthetic and natural colorants that will immobilize the gelatin-based film. Therefore, the gelatin-based film color indicator can be utilized as an effective tool to monitor and control the shelf-life of packaged foods, especially meat products, to optimize distribution, manage the stock rotation system, and reduce food waste.

## Figures and Tables

**Figure 1 foods-11-03797-f001:**
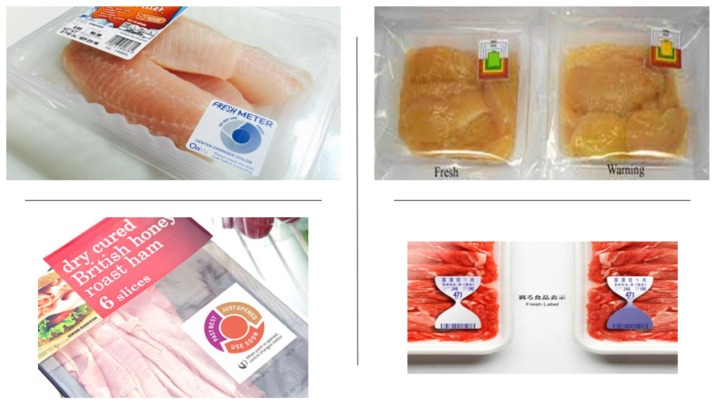
Various types of color indicators in food packaging [45].

**Figure 2 foods-11-03797-f002:**
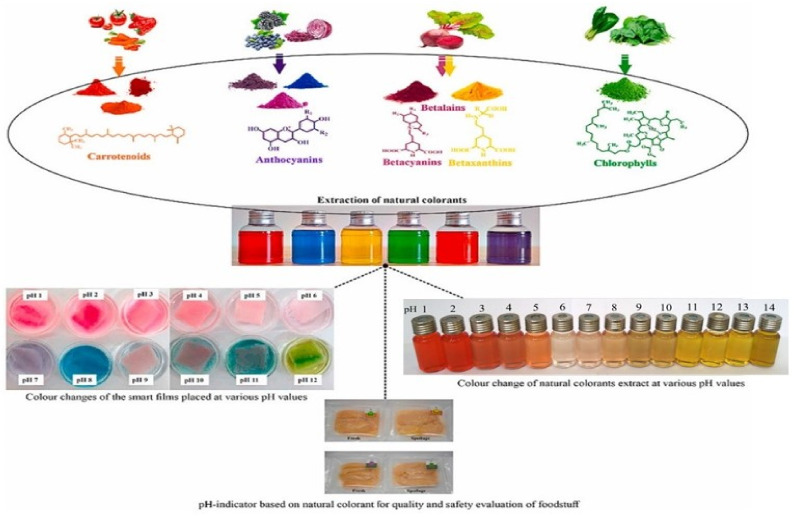
The color difference of natural colorants in various levels of pH range [53].

**Table 1 foods-11-03797-t001:** Effect of color stability using various sources of natural extract for the application of film packaging as a color indicator.

Bioactive Compound	Source	Concentration (*w*/*v*%)	Treatment			Color Difference	Reference
			Days	Temperature (°C)	Condition		
Anthocyanin	Black bean	2	28	25	Light	Change	[6]
	Red cabbage	2	28	25	Light	Stable	
Anthocyanin	Red radish	2	14	25	Light	Change	[8]
		2	14	4	Light	Stable	
Anthocyanin (ATH); curcumin (CR)	Purple sweet potato; turmeric	(ATH) 8:2 (CR)	180	25	Light	Change (high)	[109]
		(ATH) 2:8 (CR)	180	25	Light	Change (low)	
Anthocyanin	Mulberry	2	24	25	Light	More stable	[110]
		2	24	25	Ultraviolet	Less stable	
Anthocyanin	Butterfly pea	0.2	16	25	Light	Change (high)	[111]
		0.2	16	4	Light	Change (low)	
Anthocyanin	Grape	1.5	60	20	Dark	More stable	[112]
		1.5	60	20	Light	Less Stable	
		1.5	60	4	Dark	More stable	
		1.5	60	4	Light	Less Stable	
Anthocyanin	Roselle	3	14	25	Light	Less stable	[48]
		3	14	4	Light	More stable	

**Table 2 foods-11-03797-t002:** pH range of film packaging as a color indicator from various sources of natural extract and application on meat spoilage observation.

Sample	Bioactive Compound	Source	pH Range	Color	Treatment		Color Changes	Reference
					Storage Period	Temperature (°C)		
Fish	Anthocyanin	Purple/sweet potato	<5.0	Red	48 h	20	Red to blue	[96]
			5.0–6.0	Pink				
			7.0	Purple				
			8.0	Blue				
			9.0–10.0	Green				
			11.0–12.0	Yellow				
		Roselle	2.0–3.0	Red	165 h	4	Red to blue	[48]
			4.0–6.0	Pink				
			7.0	Colorless				
			8.0–9.0	Blue				
			10.0–12.0	Yellow green				
		Mulberry	2.0–3.0	Bright Red	36 h	35	Bright red to dark green	[117]
			4.0–6.0	Purple				
			7.0–8.0	Pale purple				
			9.0–11.0	Dark green				
	Curcumin	Turmeric	6.0	Yellow	8 days	4	Yellow to dark red	[118]
			>6.0	Red				
Shrimp	Anthocyanin	Purple potato	<5.0	Red	48 h	25	Red to blue	[97]
			5.0–6.0	Pink				
			7.0	Purple				
			8.0	Blue				
			9.0–10.0	Green				
			11.0–12.0	Yellow				
		Purple sweet potato	<5.0	Red	18 h	25	Red to blue	[115]
			5.0–6.0	Pink				
			7.0	Purple				
			8.0	Blue				
			9.0–10.0	Green				
			11.0–12.0	Yellow				
		Red radish	2.0	Orange	48 h	30	Red to purple	[8]
			3.0–4.0	Pink				
			5.0–7.0	Pink purple				
			8.0–9.0	Purple				
			10.0	Blue				
			11.0–12.0	Yellow				
	Curcumin	Turmeric	6.0	Yellow	36 h	4	Yellow to orange red	[119]
			>6.0	Red				
Chicken	Anthocyanin	Bilberry	2.0–4.0	Red	4 days	25	Red to blue	[120]
			5.0–6.0	Purple				
			7.0–9.0	Blue				
			10.0–11.0	Green				
			12.0	Yellow				
	Curcumin	Turmeric	6.0	Yellow	8 days	4	Yellow to orange	[121]
			>6.0	Red				
Ground beef	Anthocyanin	Purple and black rice	3.0	Red	2 days	25	Red to blue	[122]
			4.0–7.0	Pink				
			8.0–9.0	Blue				
			>9.0	Purple				
Pork	Anthocyanin	*Clitoria ternatea*	1.0–3.0	Red	48 h	25	Pink/purple to green	[50]
			4.0–5.0	Purple				
			6.0–7.0	Blue				
			8.0–9.0	Green				
			10.0–11.0	Colorless				
			12.0	Yellow				
		Raspberry pomace	1.0–3.0	Red	12 h	25	Bright red to blue	[110]
			4.0–6.0	Pink				
			7.0 10.0	Blue purple				
			11.0–13.0	Green				

**Table 3 foods-11-03797-t003:** Absorption peak of the film packaging as a color indicator at different concentrations of extract.

Bioactive Compound	Source	Concentration (%*w*/*v*)	Fourier Transform Infrared (FTIR) Absorption Peak (cm^−1^)					Reference
			-OH	-CH	C-O	C=O	C=C	
Anthocyanin	Mulberry pomace extract	0	3329	2930	1034	1724	1609	[119]
		10 (free)	3327	2928	1034	1724	1606	
		20 (free)	3331	2930	1034	1724	1608	
		10 (microencapsulated)	3318	2932	1033	1724	1608	
		20 (microencapsulated)	3320	2924	1033	1720	1608	
Anthocyanin	Purple rice	0	3248	2924	1151	1634	1543	[112]
		1	3250	2925	1151	1634	1541	
		3	3249	2926	1151	1634	1543	
		5	3255	2924	1151	1633	1546	
	Black rice	0	3248	2924	1151	1634	1543	
		1	3255	2922	1152	1633	1547	
		3	3248	2923	1152	1633	1546	
		5	3249	2923	1152	1633	1546	
Anthocyanin	Purple corn	0	3256	2924	1152	1633	1547	[124]
		2	3262	2929	1152	1635	1545	
		2 + silver nanoparticle	3253	2925	1152	1634	1544	
Anthocyanin	Haskap berries	0	3284	2934	1109	1634	1534	[123]
		0.5	3284	2923	1109	1633	1534	
		1	3284	2923	1109	1633	1534	
		2	3284	2923	1109	1633	1534	
		3	3284	2923	1109	1632	1533	
Betalains	Cactus pears	0	3292	2922	Not stated	1650	1562	[100]
		1	3295	2927		1651	1562	
		2	3293	2933		1650	1562	
		3	3294	2932		1650	1563	
Anthocyanin	*Lycium* *ruthenicum*	0	3326	2928	1149	1641	Not stated	[124]
		1	3327	2928	1149	1641		
		2	3320	2928	1149	1642		
		4	3319	2928	1149	1642		
Anthocyanin	Mulberry	0	3349	2923	1159	1642	Not stated	[16]
		1	3358	2937	1159	1643		
		2	3347	2936	1159	1643		
		4	3339	2936	1159	1643		

**Table 4 foods-11-03797-t004:** Relationship between the concentration of natural extract and water-vapor permeability of gelatin-based film.

Natural Extract	Source	Concentration (wt%)	Water Vapor Permeability (×10^−u^ g m^−1^ s^−1^ Pa^−1^)	Effect on Film	Reference
Anthocyanin	Haskap berry extract	0.5	7.14 ± 0.64	Decreased gradually with increasing of the extract concentration. More complex network is formed. Decrease in amorphous region that leads to low transferable of water molecule.	[123]
		1.0	6.99 ± 0.33		
		2.0	6.27 ± 0.53		
		3.0	5.96 ± 0.21		
Anthocyanin	Mulberry pomace extract	1.0	4.24 ± 0.24	Decreased gradually with increasing of the extract concentration. Compact and dense network formed through intermolecular interactions between the extract and the film.	[17]
		2.0	4.08 ± 0.13		
		4.0	3.86 ± 0.12		
Anthocyanin	Butterfly pea	0	0.76 ± 0.08	Decreased in the presence of the extract.	[128]
		Not stated	0.73 ± 0.14		
Anthocyanin	Red cabbage	0	6.80 ± 0.40	Decreased in the presence of the extract. Lowered the cross-linking of the protein.	[129]
		Not stated	6.50 ± 0.60		
Anthocyanin	Saffron petals	0	2.46 ± 0.05	Decreased in the presence of the extract. Reduced the pore size and enhanced tortuosity of the gelatin-based film Retarded the molecular diffusion of water molecules through the film.	[130]
		Not stated	2.23 ± 0.08		
	Red barberry	0	2.46 ± 0.05		
		Not stated	2.22 ± 0.05		
Curcumin	Turmeric	0	6.50 ± 0.30	Decreased in the presence of the extract.	[99]
		0.02	1.00 ± 0.01		
Curcumin	Turmeric	0	4.12 ± 0.65	Decreased in the presence of the extract. Increased the hydrophobic character of the gelatin-based film.	[118]
		Not stated	3.65 ± 0.23		
Curcumin	Turmeric	1.0	2.14 ± 0.07	Decreased gradually with increasing of the extract concentration. Gelatin polymer contains polar molecules that can interact with water molecule that leads to the immobilization of water vapor to pass through the film.	[116]
		2.0	2.12 ± 0.08		
		3.0	1.52 ± 0.02		
		4.0	1.53 ± 0.02		

**Table 5 foods-11-03797-t005:** Effect of natural extract on antimicrobial and antioxidant properties of film packaging as a color indicator.

Bioactive Compound	Source	Concentration (%)	Diameter of Inhibition (mm)		Antioxidant Properties (DPPH Radical Scavenging Activity)	Reference
			Gram (+ve)	Gram (−ve)		
Catechin	Green tea extract	0	*S. aureus* 0	*E. coli* 0	Not stated	[134]
		5	14.5	0	42.53%	
		0	18.5	16.0	49.21%	
		20	22.5	16.0	43.92%	
	Pu-erh tea extract	0	*S. aureus* 0	*E. coli* 0	The antioxidant properties increased with increasing extract concentration	
		5	0	0		
		10	0	0		
		20	16.0	0		
Anthocyanin	Purple corn extract	0	*S. aureus* 1.69 ± 0.11	*E. coli* 1.30 ± 0.01	The antioxidant properties increased with increasing extract concentration	[124]
		2	6.57 ± 0.04	5.40 ± 0.31		
		0	*L. monocytogenes* 1.39 ± 0.01	*Salmonella* 1.67 ± 0.18		
		2	5.48 ± 0.25	5.92 ± 0.20		
Betalains	Cactus pear extract	0	*L. monocytogenes* 11.1 ± 0.3	*E. coli* 10.4 ± 0.1	The antioxidant properties increased with increasing extract concentration	[100]
		1	12.5 ± 0.1	10.4 ± 0.2		
		2	13.1 ± 0.9	10.5 ± 0.1		
		3	12.8 ± 0.4	10.3 ± 0.2		
Anthocyanin	Red barberry	0	*S. aureus* 0	*E. coli* 0	Not stated	[135]
		3	16.8 ± 2.2	18.3 ± 0.34	82.1%	
Anthocyanin	Saffron petal	0	*S. aureus* 0	*E. coli* 0	The antioxidant properties increased with increasing extract concentration	[114]
		3	22.8 ± 1.5	20.2 ± 3.3		

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
