# Peer review of "Gelatin-Based Film as a Color Indicator in Food-Spoilage Observation: A Review"

_foods, 2022, doi:10.3390/foods11233797_

Round 1
Reviewer 1 Report
Usually, reviews are quite useful to the scientific progress of a particular research field. Reviews provide readers with a useful over-view of knowledge about the focal phenomenon, as well as insights on key dilemmas and conflicting findings that suggest future research directions. In this work the authors have present a review of a current topic of interest in the world of smart packaging materials development.
Nevertheless, the revised manuscript lacks important information that could limit the usefulness of the work. From my point of view, one of the main goals of a revision should be to ensure its replicability by other researchers, hence information about how the collection of articles that are reviewed was gathered, i.e., how the process of the search for articles was performed and the article inclusion (or article exclusion) criteria as well needs to be provided. Probably with this information some exclusions could be explained, for example (just an illustration):
-Gelatin based films capable of modifying its color against environmental pH changes (by Musso et al.)
-Aqueous hibiscus extract as a potential natural pH indicator incorporated in natural polymeric films (by Peralta et al.)
-Preparation and characterization of a novel colorimetric indicator film based on gelatin/polyvinyl alcohol incorporating mulberry anthocyanin extracts for monitoring fish freshness (by Chen et al.)
I will recommend authors to add their search strategy.
The manuscript needs a serious revision of the English. Just an example extracted from lines 188 to 193: “There is various research conducted by previous researchers who found specific microorganisms that induce meat spoilage in poultry meat, is Pseudomonas, Enterobacteriace, lactic acid bacteria, and Brochothrix thermosphacta [33,34], lamb meat is Pseudomonas, Shewanella putrefaciens, Brochothrix thermosphacta, Enterobacteriaceae, lactic acid bacteria, and Clostridium spp. [35], beef meat is carnobacterium, Brochothrix, and Leuconostoc spp”. Please review and improve the English throughout the entire document, potential readers will acknowledge the effort made.
The Food Spoilage section needs to be shortened, as most of the information in this section does not contribute to the main goal of the review.
A section summarizing the different approaches used to extract and/or isolate the natural compounds used as indicators could add extra value to the review.
Figure 1 shows examples of food packaging containing colour indicator systems, are these gelatine-based films?
What is the actual situation of gelatine-based films in the packaging market? A short paragraph including this information in the introduction section will add more interest to the review.
Author Response
Dear reviewer,
The authors appreciate the reviewer very much for your work during the review of this manuscript. The comments from reviewers have been considered, and the manuscript has been revised accordingly.
Best wishes,
Norizah.
UMT, Malaysia.

Reviewer 2 Report
1-At reference 116 in Table 1 (Line 557), the source is unclear. Is it potatoes?
2-In the summary part, first of all, the subject of the compilation should be summarized. Then, at the end of the summary, unlike other compilation studies, it would be more elegant to finish the summary by stating what you compile with this compilation study.
3-The following information in the abstract at the end of the introduction could have replaced the review summary.
‘Active packaging is usually used to extend the shelf life of food products, as well as to improve the safety and maintain the quality of the food product, while smart packaging has the ability to monitor the packaged food and indicate the product’s quality and freshness information for manufacturers, retailers, and consumers [12]. Commonly, the biopolymers used to immobilize the color pigment in functioning as smart packagings are protein-based film (chitosan, gelatine, and agar) and polysaccharide-based film (starch, carboxymethyl cellulose, and konjac glucoman nan) [11]. Gelatine is a good biopolymer for film formation due to its film-forming ability, biocompatibility, and biodegradability, which makes it suitable to be applied to food packaging [13]. The gelatine-based film has been proven to provide good protection against aroma, oxygen, and light, although it has some major drawbacks, such as poor barrier properties against water vapor and being highly sensitive to moisture [14]. This review would be able to provide alternative suggestions to solve the food spoilage determination especially for perishable food by giving adequate exposure to the usage of the color indicators from the gelatine-based film.’ Line 63-88
4-That sentence seems strange to me. Looking at the more general literature instead, I don't know if an expression like this would be more appropriate. Line 88
As far as the authors are aware …
5-Food spoilage occurs during storage. Are they all of microbiological origin? If the color of a food item has changed for different reasons, this change has been seen before the package has been opened, should that food be thrown away? Information transitions with answers to such questions would be interesting.
6-Looking at the literature, I have not come across a compilation about color, as the authors have stated, and I would like to state that it is a good study in that respect. The compilation study is mainly based on the characteristics of natural resources used as color indicators.
7-The following statement given in Chapter 8 could have been included in the summary and introduction.
‘Scientists have working on alternative ways to develop color indicators in detecting food spoilage. The color indicator is able to detect food spoilage due to its sensitive nature toward the various level of pH by incorporating colorants from synthetic or natural sources as well as able to detect gas levels such as carbon dioxide, ammonia, and oxygen in food spoilage observation.’ Line 739-743
8-Prion issue is also important in food safety. Gelatin is a sensitive issue. Choosing fish as a source of gelatin does not eliminate the danger of prions. Such studies can also be added.
https://www.ncbi.nlm.nih.gov/pmc/articles/PMC7766531/
https://efsa.onlinelibrary.wiley.com/doi/full/10.2903/j.efsa.2020.6267
9-In the statement at the end of the conclusion section, a generalization was made by saying color indicators, but only gelatin-based color indicator was mentioned in the title. Line 765
10-Gelatin is an animal product, and scientists can change its color indicator property by adding color substances of vegetable origin to it. So it would be clearer if the whole article focused on that. It's in the tables. It would be better if the times given in the tables were explained more clearly.
Author Response
Dear reviewer,
The authors appreciate the reviewer very much for your work during the review of this manuscript. The comments from reviewers have been considered, and the manuscript has been revised accordingly.
Best wishes,
Norizah,
UMT, Malaysia.

Round 2
Reviewer 1 Report
The authors have addressed the comments and questions of the review provided.